# The Critical Role of Hypoxia in the Re-Differentiation of Human Articular Chondrocytes

**DOI:** 10.3390/cells11162553

**Published:** 2022-08-17

**Authors:** Carlos Martinez-Armenta, Carlos Suarez-Ahedo, Anell Olivos-Meza, María C. Camacho-Rea, Laura E. Martínez-Gómez, Guadalupe Elizabeth Jimenez-Gutierrez, Gabriela A. Martínez-Nava, Luis E. Gomez-Quiroz, Carlos Pineda, Alberto López-Reyes

**Affiliations:** 1Graduate Program in Experimental Biology, DCBS, Universidad Autónoma Metropolitana Iztapalapa, Mexico City 09340, Mexico; 2Laboratorio de Gerociencias, Instituto Nacional de Rehabilitación “Luis Guillermo Ibarra Ibara”, Mexico City 14389, Mexico; 3Departamento de Reconstrucción Articular, Instituto Nacional de Rehabilitación “Luis Guillermo Ibarra Ibara”, Mexico City 14389, Mexico; 4Servicio de Medicina del Deporte y Ortopedia, Instituto Nacional de Rehabilitación “Luis Guillermo Ibarra Ibara”, Mexico City 14389, Mexico; 5Departamento de Nutrición Animal, Instituto Nacional de Ciencias Médicas y Nutrición Salvador Zubirán, Secretaria de Salud, Mexico City 14080, Mexico; 6Laboratorio de Medicina Experimental y Carcinogénesis, Departamento de Ciencias de la Salud, Universidad Autónoma Metropolitana Iztapalapa, Mexico City 09340, Mexico; 7Dirección General, Instituto Nacional de Rehabilitación “Luis Guillermo Ibarra Ibara”, Mexico City 14389, Mexico

**Keywords:** chondrocyte, hypoxia, cartilage, re-differentiation, 3D cell culture

## Abstract

The preservation of the chondrogenic phenotype and hypoxia-related physiological microenvironment are major challenges in the 2D culture of primary human chondrocytes. To address this problem, we develop a 3D culture system generating scaffold-free spheroids from human chondrocytes. Our results highlight the chondrogenic potential of cultured human articular chondrocytes in a 3D system combined with hypoxia independently of the cartilage source. After 14 days of culture, we developed spheroids with homogenous diameter and shape from hyaline cartilage donors. Spheroids generated in hypoxia showed a significantly increased glycosaminoglycans synthesis and up-regulated the expression of *SOX9*, *ACAN*, *COL2A1*, *COMP*, and *SNAI1* compared to those obtained under normoxic conditions. Therefore, we conclude that spheroids developed under hypoxic conditions modulate the expression of chondrogenesis-related genes and native tissue features better than 2D cultures. Thus, this scaffold-free 3D culture system represents a novel in vitro model that can be used for cartilage biology research.

## 1. Introduction

Osteoarthritis (OA) is a degenerative joint disease associated with cartilage damage, synovitis, and altered bone remodeling [1]. Globally, OA is the most prevalent chronic joint disease, causing motor disability in the elderly where the quality of life is detrimentally affected [2,3]. Moreover, its adverse impact on health systems and economies will persist with the worldwide aging population. For decades, there has been a relentless struggle to find bioactive molecules for cartilage tissue engineering and to treat the early stages of OA that could decrease its progression in the long term. However, one of the challenges we face is designing in vitro models that mimic intra-articular conditions, making the translation of new drugs to humans highly complex.

Three-dimensional (3D) culture of chondrocytes and hypoxic conditions has been proposed as novel approaches to assay cartilage repair treatments [4,5,6,7]. These insightful culture methods provide an attractive option to mimic the tissue microenvironment since they can establish the cell–cell and cell–matrix interactions required for maintaining chondrocyte function and phenotype frequently lost in the monolayer culture [8,9,10]. Nevertheless, establishing an accurate 3D culture system using human de-differentiated articular chondrocytes as starting material represents a challenge due to cartilage source limitation and the marked de-differentiation of human chondrocytes as a result of in vitro expansion required to reach confluency [11,12].

Furthermore, the hypoxic culture condition has been described as a critical factor in emulating the cartilage microenvironment retaining chondrocyte phenotype and reducing cell de-differentiation [4]. Moreover, several studies have also evaluated hypoxia as an innovative culture platform to induce chondrogenic differentiation of mesenchymal stem cells [13,14,15]. In this sense, human chondrocyte culture in 3D systems combined with hypoxia could encourage a sustainable expression of cartilage-specific genes, supporting the re-differentiation of de-differentiated chondrocytes to assemble the native 3D structure of cartilage. 

This work represents an innovative method to develop a hypoxic 3D culture system of human osteoarthritic chondrocytes that modulates the expression of chondrogenic genes. The cell culture under hypoxia allows us to use articular chondrocytes derived from osteoarthritic donors as starting material for generating scaffold-free spheroids with native tissue features required for cartilage biology research.

## 2. Materials and Methods

### 2.1. Human Articular Chondrocytes

Hyaline cartilage biopsies (macroscopically healthy) were obtained from a non-weight bearing area of the knees from patients who underwent either knee arthroscopy or total knee replacement. This study was conducted following the Declaration of Helsinki and approved by the Ethics and Research Committee of the Instituto Nacional de Rehabilitación Luis Guillermo Ibarra Ibarra (39/20 SP-1). All participants were appropriately informed about the study’s aim and signed the informed consent form. Cartilage samples from 15 donors were included in this study: four anterior cruciate ligament (ACL) reconstructions, four partial meniscectomies, one lateral meniscus fixation, and six total knee replacements. All patients underwent surgery in the usual manner and received the same type of anesthesia (spinal with sedation).

### 2.2. Primary Isolation and Culture of Human Chondrocytes

Immediately after the harvesting process, cartilage samples were subjected to mechano-enzymatic breakdown. Cartilage samples were cut into small pieces. Chondrocytes were isolated by digestion with 2% type II collagenase (Gibco, Thermo-Fischer Scientific, Waltham, MA, USA) for 6 h in an incubator at 37 °C under constant stirring (Enviro-Genie, Scientific Industries Inc., Bohemia, NY, USA) [16]. Isolated cells were cultured in DMEM/F12 medium supplemented with 10% fetal bovine serum (FBS) and 1% antibiotics (Gibco, Thermo-Fischer Scientific, Waltham, MA, USA). Cell cultures were maintained at 37 °C with 5% O_2_ and 5% CO_2_ at saturation humidity (hypoxia) or 37 °C with 21% O_2_ and 5% CO_2_ at saturation humidity (normoxia). Chondrocytes from passage 2 (P2) were used to generate the spheroids.

### 2.3. Estimating the De-Differentiation Rate of Human Articular Chondrocytes in Monolayer Culture

The percentage of cells with a chondrocyte-like morphology was analyzed by considering the shape and diameter previously described [17,18]. Briefly, images from cells cultured under hypoxia or normoxia were taken with light-microscopy using an inverted microscope (EVOS FL Auto, Thermo-Fischer Scientific, Waltham, MA, USA). A total of 100 cells were counted in three independent events.

### 2.4. Manufacturing of Non-Adhesive Microwells

Ultrapure agarose was dissolved in PBS (4% *w*/*v*), heated, and poured on the bottom of a 12-well plate to stamp negative polydimethylsiloxane (PDMS) customized mold (Research Micro Stamps, Clemson, SC, USA) containing circular pillar structures. Molds with pores of 250 μm diameter and a depth of 250 μm were generated. Once the agarose polymerized at room temperature, the agarose microwell was separated from the PDMS mold. Subsequently, each 12-well plate was collocated under UV radiation for 60 min and then stored at 4 °C until use.

### 2.5. Formation of Human Cartilage Spheroids

Spheroids were formed by seeding 6 × 10^5^ chondrocytes in agarose microwell in a chondrogenic medium consisting of DMEM-F12 supplemented with 5% FBS, 50 mg mL^−1^ ascorbic acid-2-phosphate (Sigma-Aldrich, St. Louis, MO, USA), 50 μL 100× Insulin-Transferrin-Selenium (Gibco, Thermo-Fischer Scientific, Waltham, MA, USA), 100 mg mL^−1^ Dexamethasone, 10 ng mL^−1^ TGF-β1 (PeproTech, Cranbury, NJ, USA), and 100 units mL^−1^ penicillin and 100 units mL^−1^ streptomycin (Gibco, Thermo-Fischer Scientific, Waltham, MA, USA). Cells were placed in a humidified incubator in hypoxic or normoxic conditions for 14 days, as stated in Section 2.2. The culture medium was changed every 48 h. The number of cells per spheroid was estimated based on the number of microwells and the number of cells seeded.

### 2.6. Live/dead Viability Assay

Spheroids in agarose microwell were washed with PBS and incubated with calcein-AM (2 μg mL^−1^) and propidium iodide (2 μg mL^−1^) (Thermo-Fischer Scientific, Waltham, MA, USA). After 30 min of incubation, viability was evaluated using an inverted fluorescence microscope and Pearl Scope 64 image software (EVOS FL Auto, Thermo-Fischer Scientific, Waltham, MA, USA). ImageJ software (NIH, Bethesda, MD, USA) was used to quantify the fluorescence signal. The mean fluorescence signal for calcein and propidium iodide was obtained.

### 2.7. Morphometric Evaluation

The morphology of spheroids was analyzed using Pearl Scope 64 image software (EVOS FL Auto, Thermo-Fischer Scientific, Waltham, MA, USA). The diameter, area (A), and perimeter (p) were measured throughout 14 days of hypoxia and normoxia culturing. Circularity was calculated using the formula f_circularity_ = (4πA)/p^2^.

### 2.8. Glycosaminoglycans Staining

Spheroids were flushed out of agarose microwell and centrifuged in a 12-well plate at 1500 rpm for 1 min. The medium was removed, and spheroids were washed with PBS and fixed in 4% formaldehyde. Staining for glycosaminoglycans (GAG) of the extracellular matrix (ECM) was performed using Alcian blue in 3% acetic acid (pH 2.5) for 40 min. The quantification of staining was performed using the ImageJ software. Color thresholding was used to select and measure blue stained areas (Alcian blue). The percentage of Alcian blue staining was calculated by dividing the blue-stained area by the total spheroid area.

### 2.9. Gene Expression Analysis

We used qRT-PCR to analyze the expression of chondrogenesis-related genes. Primary human chondrocytes and spheroids were dissolved in 0.6 mL Lysis Buffer (Pure Link RNA, Thermo-Fischer Scientific, Waltham, MA, USA), and total RNA was isolated according to the manufacturer’s protocol. A total RNA (500 ng) was reverse transcribed in a 20 μL volume reaction using High-Capacity cDNA Reverse Transcription Kit (Applied Biosystems, Thermo-Fischer Scientific, Waltham, MA, USA). cDNA products were stored at −20 °C until used. The qRT-PCR was carried out using 10 μL of RT2 SYBR Green FAST Mastermix (Qiagen, Germany) in a final volume of 20 μL under the following condition: 95 °C for 10 min, 40 cycles at 95 °C for 15 s, and variable Tm and time for annealing and elongation step (Table 1). The qRT-PCR amplification was performed by the Rotor-Gene Q thermocycler (Qiagen, Germany).

### 2.10. Statistical Analysis

The normality of the input data was tested with the Shapiro–Wilk normality test. For spheroid diameter and Alcian blue staining of hypoxic spheroids, one-way analysis of variance (ANOVA) followed by the Tukey post hoc test was performed to compare multiple groups. For the analysis of the percentage of cells with a chondrocyte-like morphology in normoxia and hypoxia throughout the culture time, we performed a two-way ANOVA test.

The student *t*-test was performed to analyze the difference between normoxia and hypoxia-related gene expression in spheroids, GAG staining, and viability assay. The Mann-Whitney U test was carried out for values that were not normally distributed. Among multiple groups was performed with the Kruskal–Wallis test followed by Dunn’s pairwise comparison as a post hoc test for gene expression in primary cell culture. Significance was assigned at *p* < 0.05. 

If not otherwise stated, data are presented as median with interquartile range (IQR) or mean with 95% confidence interval (CI). The statistical analysis was carried out using the STATA v.13 statistical package (StataCorp, College Station TX, USA). Graphics were generated using GraphPad Prism 9.0.1 (GraphPad Software, San Diego CA, USA).

## 3. Results

### 3.1. Anthropometric Features of Cartilage Donors

We analyze the impact of age, body mass index, gender, and procedure stratified by normoxic or hypoxic culture. There was no significant difference in age (*p* = 0.16), body mass index (*p* = 0.33), gender (*p* = 0.59), and procedure (*p* = 0.71) (Table 2). Macroscopically, cartilage samples showed a smooth and intact surface.

### 3.2. De-Differentiation of Articular Chondrocytes in Traditional 2D Cell Culture

#### 3.2.1. Morphology of Passaged Articular Chondrocytes

Human articular chondrocytes from cartilage biopsy were cultivated in hypoxic or normoxic conditions until confluence and designated as passage 0 (P0). The articular chondrocytes showed a rounded, polygonal morphology at P0 (Figure 1a,b). However, there was a progressive change to fibroblast-like morphology through the passages, mainly observed in chondrocytes from normoxic culturing conditions (Figure 1b). Chondrocytes at P2 still exhibited a native chondrocyte morphology when cultured in a low oxygen tension (5% O_2_) incubator (Figure 1c). Significant differences in the number of cells with chondrocyte-like morphology were mainly observed at P0 and P1 between normoxia and hypoxia. Moreover, articular chondrocytes cultured in hypoxia presented a higher growth rate by reaching confluency on average after 25 days versus 31 days required in normoxia.

#### 3.2.2. Downregulation of Chondrogenic Genes in Cultured Chondrocytes

Gene expression analysis of passaged articular chondrocytes revealed a slight tendency to the de-differentiation process. Changes in gene expression were marked in normoxic culture conditions as *COL2A1* levels significantly differed between P0 (−2.08 IQR (−2.54, −1.11)) and P1 (−2.80 IQR (−3.16, −2.47)) (*p* = 0.021) as well as P0 vs. P2 (−2.78 IQR (−2.97, −2.41)) (*p* = 0.032) (Figure 1d). During in vitro cultivation in hypoxia, chondrocytes showed a decreased expression of *COL2A1* only from P0 (−1.58 IQR (−1.79, −0.64)) to P1 (−2.66 IQR (−2.84, −2.03)) (*p* = 0.022). Regarding *SOX9* and *ACAN* expression, non-significant differences were found in monolayer culture under normoxia and hypoxia.

There was a stable expression of the fibroblast marker *COL1A1* throughout passages in normoxia and hypoxia. Nevertheless, the expression of *COL10A1*, a hypertrophic marker, showed decreased levels with increasing passage number in hypoxia (P0 vs. P2, *p* = 0.013) (Figure 1e). In contrast, normoxia exerted higher expression levels of *COL10A1* toward in vitro expansion, being statically significant at P2 (*p* = 0.038). Comparing the relative expression of *COL2A1*/*COL1A1*, there was a 1.7-fold increase at P0 in hypoxia (−1.81 (95% CI −2.53, −1.08)) versus normoxia (−3.15 (95% CI −3.81, −2.49)) (*p* = 0.007). Nevertheless, the *COL2A1*/*COL1A1* ratio was clearly reduced through passages in normoxia. On the other hand, the ratio remained higher in chondrocytes cultured in hypoxia at P2 (−2.88 (95% CI −3.62, −2.14)) than those under normoxic conditions (−3.76 (95% CI −4.10, −3.43)) (*p* = 0.020).

Of particular significance, we observed in hypoxic culturing conditions at P2 the highest *COL2A1*/*COL10A1* ratio of 3.25 ((95% CI 2.66, 3.83.) *p* = 0.012) in comparison to normoxia (2.41 (95% CI 2.07, 2.74)). In contrast to hypoxia, the gene expression ratio of *COL2A1*/*COL10A1* in chondrocytes cultured in normoxia was reduced with increasing culture time.

### 3.3. Developing Scaffold-Free Spheroids from Human De-Differentiated Chondrocytes

#### 3.3.1. Spheroid Morphology

Cells aggregated spontaneously in agarose microwells, and regular-shaped spheroids were formed, maintaining stable circularity throughout culturing. After one day in hypoxia, compact spheroids with smooth surfaces were observed; nevertheless, spheroids generated in normoxia showed irregular borders making circularity less stable (Figure 2). Moreover, a steady size throughout the culture was kept until day 14 with hypoxic conditions (143 μm (95% CI 139–145)). In contrast, spheroids became rounded and compact with increasing culture time in normoxia, showing a final diameter of 124 μm (95% CI 120–129) on day 14 (Figure 2).

#### 3.3.2. Viability

We estimated the viability of cell-forming spheroids by live/dead assay. After 14 days of hypoxic culturing conditions, the spheroids exhibited minimal area of dead cells conserving their metabolic activity (Figure 3a). Nevertheless, a significant reduction of cell viability was detected predominantly in spheroids from normoxia, showing a low calcein signal (78.85% (95% CI 75.27, 82.44)), but not in spheroids cultured in hypoxia (97.49% (95% CI 96.52, 98.46)) (*p* < 0.001) (Figure 3b). Calcein and propidium iodide intensities were normalized to the spheroid area and expressed as a percentage of the relative intensity of fluorescence.

#### 3.3.3. Extracellular Matrix Production

The production and distribution of GAGs into spheroids were analyzed. On day 14, Alcian blue staining showed broadly distributed GAGs throughout the area in spheroids cultured in a hypoxic environment (Figure 4a). To quantify differences in GAG distribution among samples, we determined the percentage of stained area concerning total area using ImageJ software. Significant differences were found in the rate of GAGs in normoxia (70.35 ± 5.97%) compared to hypoxia (91.93 ± 2.08%) (*p* < 0.001). In this context, GAGs distribution was also conserved independently of the sample source when spheroids were generated under hypoxic conditions (Figure 4b). Interestingly, geometric features of spheroids cultured in hypoxia were preserved independently of the sample source (Figure 4c,d). A total of 10 human cartilage samples from different donors were analyzed, showing a homogeneity in size and circularity after 14 days of cell culture with a chondrogenic medium under hypoxic conditions.

#### 3.3.4. Gene Expression Profile

Re-differentiation of chondrocytes forming spheroids in normoxia and hypoxia was assessed to identify the impact of these culturing conditions on the regulation of chondrogenic gene expression. Regarding spheroids generated in normoxia, there was a significant difference in the expression ratio of *SOX9* (2.2-fold increase, *p* < 0.001) and *COL2A1* (1.6-fold increase, *p* < 0.019), taking P2 as the reference group; meanwhile, *ACAN* expression did not change drastically. Moreover, it was observed a higher *COL1A1* and *COL10A1* expression of 5.2-fold increase (*p* < 0.001) and 9484-fold increase (*p* < 0.001), respectively.

In contrast, *SOX9*, *ACAN*, and *COL2A1* were significantly upregulated in spheroids generated in hypoxia with an expression ratio of 6.8 (*p* < 0.001), 257.2 (*p* < 0.001) and 38.1 (*p* < 0.001), respectively, versus monolayer culture. Although hypoxia displayed a high expression ratio for *COL1A1* and *COL10A1*, their levels were lower than spheroids cultured in normoxia (Figure 5a).

We further compare the relative expression of different genes in spheroids after 14 days of culture in hypoxia or normoxia. Spheroids under hypoxic conditions expressed *SOX9* (*p* = 0.006), *ACAN* (*p* = 0.005), *COL2A1* (*p* < 0.001), *COMP* (*p* = 0.015), *SNAI1* (*p* = 0.014), and *LOXL2* (*p* = 0.045) at higher rates than spheroids generated in normoxia (Figure 5b). Furthermore, the relative expression of *COL1A1* appears to be significantly increased in normoxia than hypoxia (1.85 ± 0.19 vs. 1.10 ± 0.27, *p* < 0.001). This tendency was also observed for *COL10A1* (−0.735 ± 0.40 vs. −1.93 ± 0.86, *p* < 0.001) (Figure 5b).

The ratio of *COL2A1*/*COL1A1* expression in spheroids was 1.78-fold higher in hypoxia than in normoxia (*p* < 0.001). Similar results were found for the *COL2A1*/*COL10A1* ratio, where normoxia showed a lower level (−1.41 (95% CI −1.79, −1.03)) compared to hypoxia (0.52 (95% CI 0.21, 0.82)) (*p* < 0.001) (Figure 5c).

## 4. Discussion

This study demonstrates the influence of 3D culture conditions combined with hypoxia maintaining the expression of chondrogenic genes and native tissue features in chondrocytes derived from human samples with different inflammatory compromises. Previous cartilage engineering and tissue repair studies have reported similar findings related to the chondrogenic potential by combining a 3D culture system and hypoxia in rat and pig chondrocytes [6,19]. Nevertheless, to our knowledge, this work represents the first report highlighting the use of articular chondrocytes derived from osteoarthritic donors as starting material for generating scaffold-free spheroids under hypoxic conditions to promote the gene expression characteristic of native cartilage.

The in vitro expansion of human chondrocytes resembling the physiological environment of cartilage is essential to test clinically used cartilage treatments and to explore potential new therapies. However, 3D cultures also represent an opportunity to evaluate the development and progression of articular pathologies as OA at early stages [20]. Nevertheless, there is no consensus on the ideal in vitro model representing particular features of OA due to different mechanisms to induce catabolic metabolism in chondrocytes. Our data support the generation of human cartilage spheroids with homogeneous GAGs distribution resembling the original native tissue using a low-cost 3D culture method.

For the generation of spheroids in our study, we used primary cells with a certain degree of de-differentiation and phenotype alteration due to the inflammatory environment from which biopsies came. Morphology changes observed in primary chondrocytes at early passages have been previously reported. The de-differentiation phenomenon occurs due to forcing cells to grow in a 2D environment where nutrient and oxygen gradients do not exist [7,21]. In this sense, chondrocytes are prone to acquire a fibroblast-like phenotype associated with an altered gene expression of *SOX9*, *COL2A1*, and *ACAN* [11,22]. This research used chondrocytes at P2 to avoid a marked cell de-differentiation in monolayer culture under normoxia and hypoxia. Our 2D culture showed only altered expression of the chondrogenic marker, *COL2A1*, being higher affected in chondrocytes cultured in normoxia (Figure 1d); this represents a major inconvenience for in vitro culture of chondrocytes as, in turn, decreased *COL2A1*/*COL1A1* ratio indicates de-differentiation. Our study observed a significant decrease in *COL2A1*/*COL1A1* ratio with sequential passaging under normoxic conditions even when *COL1A1* expression was not altered (Figure 1d). Not surprisingly, *COL2A1*/*COL1A1* ratio was stable in chondrocytes cultured under hypoxia as *COL1A1* showed lower expression and *COL2A1* was less affected (Figure 1e).

It is worth mentioning that allowing chondrocytes to proliferate until reaching confluence may explain the steady RNA-level expression of *SOX9* and *ACAN* observed in cells at P0 vs. passaged chondrocytes from hypoxia and normoxia in our study. Nevertheless, further studies have reported that chondrocyte marker genes decreased some orders of magnitude compared to freshly isolated chondrocytes from cells at first passage [19,23]. However, it has been noticed a changeless expression of chondrogenic genes (e.g., *SOX9* and *ACAN*) in chondrocytes from human donors and animal specimens at early passages [19,21,23]. Marked changes in *SOX9* expression were observed for up to the sixth passage, while *COL2A1* declined progressively at the first cell passages of human articular chondrocytes leading to a progressive loss of the chondrogenic phenotype [21,23].

On the other hand, prolonged monolayer culture triggers the development of fibrous cartilage, which is highly undesirable because it exerts articular homeostasis imbalance [24,25,26]. Chondrocytes cultured in hypoxic conditions showed a significant decreasing trend in the expression of *COL10A1*, a marker of chondrocyte hypertrophy; nevertheless, normoxia promoted a higher expression resulting in an altered *COL2A1*/*COL10A1* ratio. These findings are consistent with previous studies reflecting de-differentiation of passaged chondrocytes under normoxia [24,27,28].

Culturing chondrocytes in a 3D environment has been widely proposed as a promising approach as it can mimic native articular signaling by re-establishing a specific phenotype and cellular functions of the cartilage compared with other previously reported methods not related to the spheroid generation [29,30,31]. Significantly, the 3D model triggers a tissue-specific phenotype due to cell–cell and cell–ECM interactions. Re-differentiation of passage cells has been demonstrated in many studies where an increased expression of *COL2A1* and *ACAN* was achieved in scaffold-based 3D culture versus 2D culture [32,33,34,35]. In our study, gene expression analysis of scaffold-free spheroids comparing monolayer culture at P2 confirmed the re-differentiation process exerted mainly under hypoxic culturing conditions. As shown in Figure 5a, *SOX9*, *ACAN*, and *COL2A1* were up-regulated in hypoxia. Nevertheless, no significant differences were observed for *ACAN* expression when spheroids were generated under normoxia. This expression profile is similar to those reported in previous studies using pellet culture and scaffold-free spheroids under hypoxia and normoxia [6,36,37,38].

It should be noted that the spheroids had an increased expression of *COL10A1*, a hypertrophic cartilage-related marker under hypoxia. High expression of hypertrophic genes has been identified during re-differentiation of chondrocytes regardless of their origin and in vitro culture method [36,39,40]. Although an elevated expression ratio for *COL10A1* was expected for normoxia, hypoxia could not deplete it (Figure 5a). However, *COL10A1* expression was significantly lowered in spheroids under hypoxic conditions than in normoxia after 14 days of culture with a chondrogenic medium (Figure 5b). It has been demonstrated in recent studies that TGF-β, found in the chondrogenic medium, can drive hypertrophic differentiation in chondrocytes while also inducing its leading role as a transforming growth factor in cartilage differentiation [41,42,43,44].

Furthermore, the differences in *COL2A1*/*COL1A1* and *COL2A1*/*COL10A1* ratio in spheroids indicate that hypoxia represents a better culturing system to control the chondrocyte features (Figure 5c). These results represent a success in our study as spheroid-forming chondrocytes maintained a genotype and function despite their osteoarthritic origin. Moreover, source limitation and time for in vitro expansion represent crucial factors that should always be considered for their implication for triggering de-differentiation in primary human chondrocytes.

Using human donor cartilage for 3D culture implies the diversification of chondrocyte phenotype obtained in diverse samples due to different OA grades observed in each patient. Regardless of the anthropometric features of donors recruited in this study, we developed size-controlled spheroids from ten hyaline cartilage using a scaffold-free system and resembling the articular environment with hypoxic culturing conditions (Figure 2). Since cartilage is a tissue with a low oxygen tension environment, it has been proposed to use an oxygen concentration of 5% for the chondrocyte in vitro culturing [45].

As the oxygen gradient has been described as an essential factor regulating cell viability dependent on spheroid size, we evaluated viability by Live/Dead assay on day 14 of human chondrocyte spheroids generated in hypoxia. In this context, our results revealed that hypoxia did not induce toxicity in the core of our in vitro 3D cartilage model (Figure 3a). Supporting our data, different studies mention that hypoxia plays a vital role in the growth and survival of mesenchymal stem cells embedded in scaffolds [46,47], being able to regulate anabolism under a hypoxic microenvironment. Although hypoxia represents the best scenario for cartilage 3D culture, adequate development of chondrocyte spheroids after 14 days of culture in normoxia combined with a chondrogenic medium was also demonstrated in this study (Figure 2). Nevertheless, spheroids from normoxic conditions showed increased dead cell areas (Figure 3a). These findings agree with a recent study where spheroids from the ATDC5 cell line were developed using a polyethylene glycol-coated chip and chondrogenic differentiation medium. The spheroids showed spotted areas of dead cells after 14 days under normoxia culturing conditions [37]. According to Omelyanenko et al., the nonviable cell population has increased significantly from days 7 to 21. Moreover, the accumulation of specific extracellular matrix components such as type II collagen and aggrecan was closely correlated with time [48], which agrees with the low GAGs synthesis observed in our spheroids under normoxia (Figure 4a).

Our results show that spheroids cultured in hypoxia showed better geometrical features and increased GAGs levels resembling the native articular cartilage than normoxia (Figure 2 and Figure 4). In this context, under hypoxic conditions, the 3D culture system produces an extensive ECM characteristic of native cartilage, suggesting that our in vitro system can regulate chondrogenic phenotype. Therefore, our in vitro 3D culture system represents a novel application for developing disease-modifying OA drugs. Our high throughput system to create human chondrocyte spheroids is based on a PDMS mold that can be easily adopted as specialized equipment is not required. Moreover, this in vitro novel tool represents an alternative for 2D cultures, facilitating the screening of active biological molecules for cartilage engineering and tissue repair, making our model easy to reproduce.

It is essential to state that further characterization of our cartilage in vitro model is required to elucidate the molecular mechanism underlying chondrogenesis in hypoxia.

## 5. Conclusions

In conclusion, we have developed scaffold-free spheroids in hypoxia from human chondrocytes derived from osteoarthritic donors, mimicking the in vivo-like cartilage microenvironment that makes possible a cartilage-specific phenotype and functions. 

This work details how hypoxia represents a critical issue in human chondrocyte culture and the combination with a 3D system to modulate the expression of chondrogenesis-related genes and native tissue features. This 3D model potentially provides a novel approach to elucidate the mechanism underlying chondrogenesis and clarify the molecular mechanism exerted by new molecules in the cartilage.

## Figures and Tables

**Figure 1 cells-11-02553-f001:**
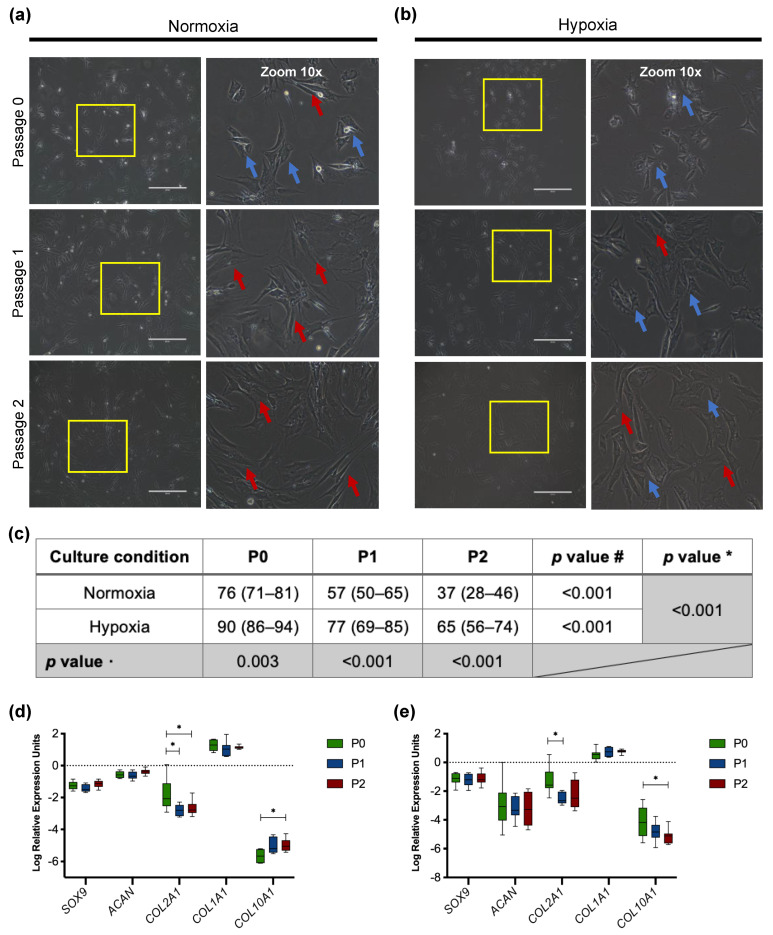
Characterization of primary cell culture of articular chondrocytes. Morphology of freshly isolated cells (P0) and passaged articular chondrocytes (P1–P2). Passaged human chondrocytes displayed features of a de-differentiated phenotype consisting of a spreading fibroblastic morphology. Blue arrows show articular chondrocyte morphology. Red arrows indicate fibroblast-like morphology. Scale bars represent 400 µm. (**a**) Chondrocytes cultured in normoxic and (**b**) hypoxic conditions. (**c**) Percentage of cells with a chondrocyte-like morphology in monolayer culture. Data are expressed as mean with 95% CI. *p*-value # for the effect of hypoxia and normoxia with increasing passage culture, *p*-value **·** for each passage between normoxia and hypoxia, and *p*-value * for two-way ANOVA. Gene expression of *SOX9*, *ACAN*, *COL2A1*, *COL1A1*, and *COL10A1* in passaged chondrocytes in (**d**) normoxia and (**e**) hypoxia. Expression normalized to the reference gene *RPL27*. Data are presented as median with IQR. Significant differences * (*p* < 0.05).

**Figure 2 cells-11-02553-f002:**
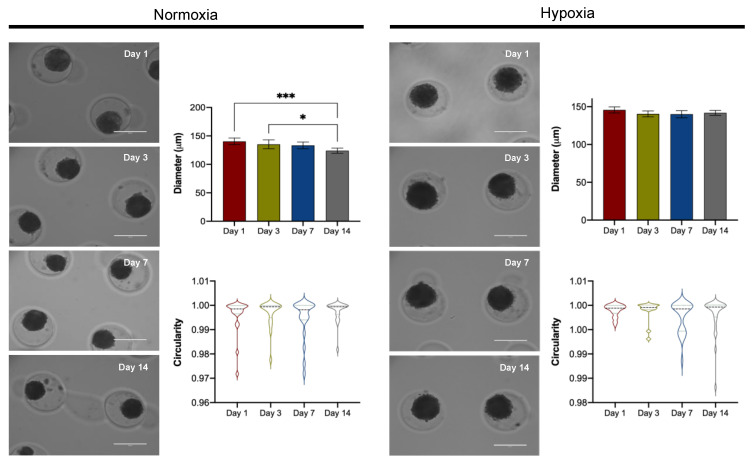
Formation of scaffold-free spheroids from human articular chondrocytes in normoxia and hypoxia over time in culture. Light microscopic images of chondrocyte spheroids on days 1, 3, 7, and 14. Geometric analysis of spheroids; diameter and circularity measurements throughout days of cell culture. Each color represents a different day. Data are presented as median with IQR for circularity and mean with 95% CI for diameter. Significant differences * (*p* < 0.05) and *** (*p* < 0.001). Scale bar = 200 μm.

**Figure 3 cells-11-02553-f003:**
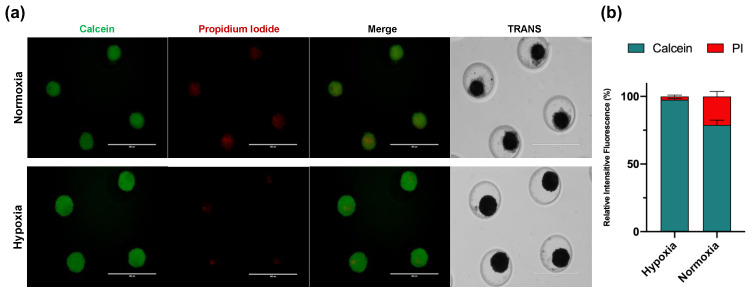
Viability assay of human chondrocytes spheroids at day 14. Viability was tested using a Calcein/Propidium iodide staining, scale bar = 400 μm. (**a**) Spheroids possessed intact morphology and many viable cells at day 14 in hypoxic culturing conditions. (**b**) Fluorescence analysis of metabolic activity and cell death in spheroids. Data are presented as the mean with a 95% CI.

**Figure 4 cells-11-02553-f004:**
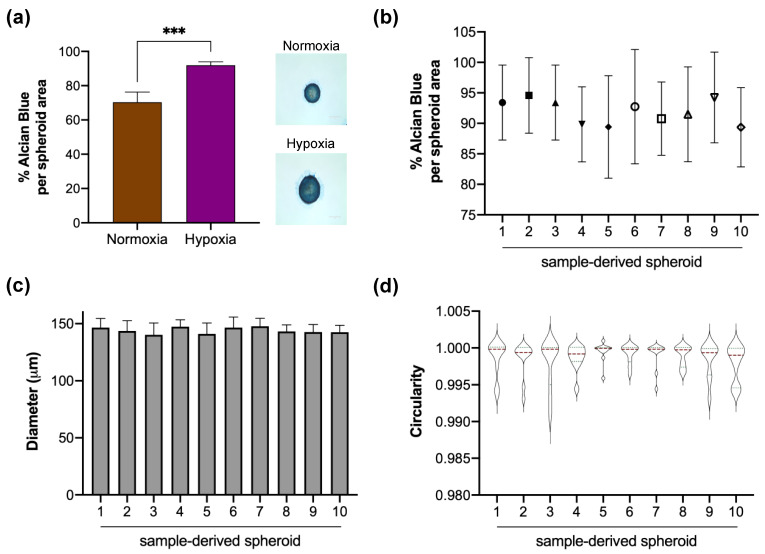
Homogeneity in geometric features among donor-derived spheroids and characterization of GAGs production in hypoxia. (**a**) Hypoxia induces higher synthesis of GAGs, a major matrix extracellular component in human cartilage spheroids. (**b**) A steady percentage of blue-stained area concerning the total area of spheroids per sample was observed under hypoxia. Spheroids in hypoxia maintained their (**c**) diameter and (**d**) circularity independently of the sample source. Scale bars represent 50 µm. Data are presented as mean ± SD for a, b and c, and median with IQR for circularity. Significant difference *** (*p* < 0.001).

**Figure 5 cells-11-02553-f005:**
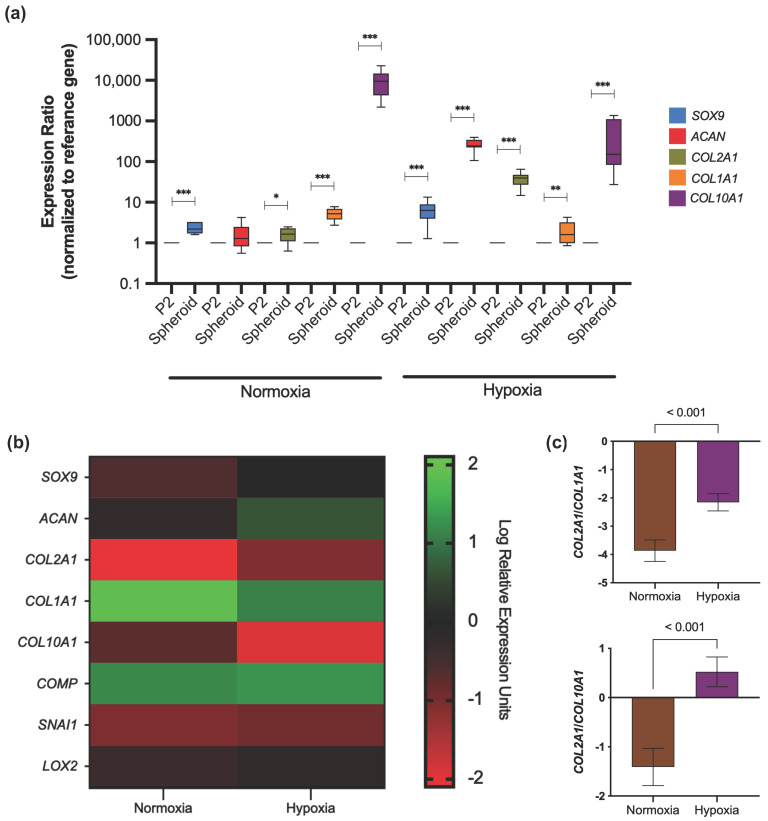
Gene expression of chondrogenesis-related genes in spheroids cultured in normoxic and hypoxic conditions. (**a**) Relative expression of chondrogenic markers in spheroids after 14 days of culturing in chondrogenic medium, taking P2 as the reference group. Expression normalized to the reference gene *RPL27*. Data are presented as median with IQR. Significant differences * (*p* < 0.05), ** (*p* < 0.01), *** (*p* < 0.001). (**b**) Log Relative Expression Units of chondrogenesis-related genes in scaffold-free spheroids generated in both culturing conditions. Expression normalized to the reference gene *RPL27*. (**c**) The ratio of *COL2A1*/*COL1A1* and *COL2A1*/*COL10A1* expression in spheroids after 14 days of hypoxic or normoxic culturing conditions. Data are presented as mean with 95% CI.

**Table 1 cells-11-02553-t001:** Primers used for gene expression analysis by qRT-PCR.

Gene Symbol	Primer Qiagen Id or Costume Sequence	qRT-PCR Condition
*SOX9*	PPH02125A	60 °C–60 seg
*ACAN*	PPH06097E	60 °C–60 seg
*COMP*	PPH07086B	60 °C–60 seg
*SNAI1*	PPH02459B	60 °C–60 seg
*LOXL2*	PPH10275A	60 °C–60 seg
*RPL27*	PPH00443B	60 °C–60 seg
*COL1A1*	PPH01299F	60 °C–60 seg
*COL2A1*	Forward ATGAGGGCGCGGTAGAGA Reverse CCCTGACACCGAAGGACAG	62 °C–47 seg
*COL10A1*	Forward CCCAGCACGCAGAATCCAT Reverse CCTGTGGGCATTTGGTATCG	58 °C–60 seg

**Table 2 cells-11-02553-t002:** Anthropometric features of cartilage donors.

Characteristics	Total (*n* = 15)	Normoxia (*n* = 5)	Hypoxia (*n* = 10)	*p*-Value
Age, years	43 ± 15	36 ± 11	47 ± 15	0.16
BMI	26.6 ± 4.1	25.1 ± 4.8	27.4 ± 3.8	0.33
Gender **				
Male	11 (73%)	4 (80%)	7 (70%)	0.59
Procedure **				
TKR	9 (60%)	3 (60%)	6 (60%)	0.71
KA	6 (40%)	2 (40%)	4 (40%)	

BMI—body mass index, TKR—total knee replacement, KA—knee arthroscopy. ** Fisher test. For age and BMI, *t*-test was performed.

## Data Availability

Not applicable.

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
