# Peer review of "The Critical Role of Hypoxia in the Re-Differentiation of Human Articular Chondrocytes"

_cells, 2022, doi:10.3390/cells11162553_

Round 1

Reviewer 1 Report

Review:

Martinez-Armenta, C et al. “The critical role of hypoxia in the re-differentiation of human articular chondrocytes”

The authors present an 3D culture system of de-differentiated human chondrocytes and applied it for hypoxic conditions to induce chondrogenic re-differentiation and proofed it by examining chondrogenesis-related genes. It is challenging to keep chondrogenic cell attributes in cell culture. Hypoxic conditions seem to further a more chondrogenic phenotype and keep fibroblast-like cells down. The paper is well-written and fairly discussed. To me there are a few points/questions:

-        Is the improvement of chondrogenic features in the hypoxic 3D-culture dependent on the size of the “chondrospheres”? In general, is the size of such a sphere limited? One could assume that bigger spheres under normoxia might also show improvements.

-        The mentioned downregulation of genes like Sox9 or ACAN in Figure 1 which probably should indicate de-differentiated chondrocytes is not convincing. A tendency does not imply a difference. Also the light-microscopy of the culture hardly shows a difference. A direct analysis of a cell type percentage having a certain proportion, shape, diameter, etc. (chondrocyte morphology) would help. An immunocytochemical/FACS analysis for chondrocyte and fibroblast markers could strongly improve the analysis.

-        In Figure 2, the legend misses indication for “a” and “b“.

In Figure 5a, the differences in the change of a gene expression are obvious but to me a direct comparison for a gene expression in the normoxic and the hypoxic spheroids would be more suitable. The P2-stage might be in general differ too much from the spheroid. There are no variances of the P2 references.

Author Response

Dear reviewer,

We appreciate your attention reviewing our manuscript and each observation made to our work improving the overall quality of our manuscript.

-        Is the improvement of chondrogenic features in the hypoxic 3D-culture dependent on the size of the “chondrospheres”? In general, is the size of such a sphere limited? One could assume that bigger spheres under normoxia might also show improvements.  
R: Regarding size of spheroids, we generated them starting from same cell density for both culture systems (600,000 cells/agarose microwell). Nevertheless, diameter was significant different in hypoxia versus normoxia. As we stated in discussion section, we consider that chondrogenic features were improved by hypoxia, which play a pivotal role promoting synthesis of major extracellular matrix components as GAGs rather than the spheroid size given that values of alcian blue staining were normalized by area (Figure 4).

-        The mentioned downregulation of genes like Sox9 or ACAN in Figure 1 which probably should indicate de-differentiated chondrocytes is not convincing. A tendency does not imply a difference. Also the light-microscopy of the culture hardly shows a difference. A direct analysis of a cell type percentage having a certain proportion, shape, diameter, etc. (chondrocyte morphology) would help. An immunocytochemical/FACS analysis for chondrocyte and fibroblast markers could strongly improve the analysis.               
R: We appreciate the insightful comment. We have now added more information in figure1 regarding percentage of cells with chondrocyte-like morphology as you recommended us for a better understanding. We would like to mention that COL2A1 was down-regulated with increasing culture passages in normoxia which affected the COL2A1/COL1A1 ratio as we stated in line 189 to 204. This result represents a clear characteristic of de-diferentiation in normoxic culture in comparison to cells under hypoxia. Moreover, we would like to mention that COL10A1, a major marker of chondrocyte hypertrophy, was up-regulated in normoxia compared to cells under hypoxia. In this sense, the ratio COL2A1/COL10A1 was clearly affected in a culture time-dependent manner in cells cultured in normoxia. Line 205 to 209.

-        In Figure 2, the legend misses indication for “a” and “b“.    
R:  We are grateful for your observation. We have added a mark for normoxic and hypoxic culture on the top of each section in figure 2.

In Figure 5a, the differences in the change of a gene expression are obvious but to me a direct comparison for a gene expression in the normoxic and the hypoxic spheroids would be more suitable. The P2-stage might be in general differ too much from the spheroid. There are no variances of the P2 references.
R: We appreciate your observation. We agree that a direct comparison for gene expression in the normoxic and the hypoxic spheroids is more suitable, which we have addressed in Figure 5 section B. Nevertheless, this section only describes the difference in gene expression of spheroids exerted by each culture condition. In this sense, we strongly believe it is important to demonstrate how the gene expression (fold change) was modulated corresponding to P2 versus its related spheroid. In that way, we can demonstrate the beneficial impact of 3D culturing. Figure 5 section A shows a higher ratio for SOX9, ACAN and COL2A1 under hypoxic conditions compared to normoxia where there was a slightly expression ratio for these cartilage-specific genes. Although COL1A1 and COL10A1 were significantly up-regulated in spheroids from normoxia than P2; in contrast, hypoxia the up-regulation of these genes was not as notorious.

We would like to mention that we could not add error bars to P2 as it was set as reference group, so its expression value was used for normalization to perform the delta-delta Ct method.

We would like to mention that we increased discussion regarding the role of hypoxia regulating gene expression in chondrocytes in 2D culture and scaffold-free spheroids. Line 323 – 331, 334 – 338, 350 – 380. We also rewrite our conclusion found in line 422 – 424.

Reviewer 2 Report

The authors described the preservation of the chondrogenic phenotype by using hypoxia-related physiological microenvironment for primary cell culture of human cartilage cells. However, it is poorly presented the purpose, rational or significance of culturing chondrocytes, either in a 2-D system or a 3D scaffold-free spheroids. As this reviewer understands, there is hardly a use of culturing chondrocytes. Also, it is not mentioned whether culture in the form scaffold-free spheroids could be scalable, assuming even if there is some uses of cultured chondrocytes. 

Author Response

Dear reviewer,

We appreciate your insightful comments to our work.

We would like to emphasize that our purpose was not to propose a novel approach to scaffold-free chondrocyte expansion for clinical applications.

We consider this work represents an innovative method to develop a hypoxic 3D culture system of human osteoarthritic chondrocytes that promotes the expression of chondrogenic genes and native tissue features required for cartilage biology research. Therefore, this in vitro 3D culture system represents a novel application for developing disease-modifying osteoarthritis drugs. Additionally, in the discussion section, we highlight that in vitro scaffold-free spheroids based on a PDMS mold could be scalable, making it easy to reproduce.

We hope this revised manuscript version increases the interest and overall quality of our work.

Reviewer 3 Report

Dear authors,

The manuscript describes hypoxic 3D culture system which promotes the redifferentiation of chondrocytes derived from human cartilage samples. The study has been done in comparison with normoxic culture. Cartilage injuries represent one of the most challenging problems in modern medicine. The scaffold free 3D culture system has been proposed to work as an efficient model for cartilage biology research. Overall, the study is systematic and well planned. However, the following concerns need to be addressed.

Major concerns:

(1) Data represented in Figure 1- SOX9, COL1A1 expression levels are similar in normoxic and hypoxic cultures. ACAN expression data shows large variation, especially for hypoxic culture-P0. In all only COL2A1 and COL10A1 have some difference in expression. Looking at the p values, this experimentation is not very conclusive.

(2) section 3.3.2 How was the viability assay performed- did the authors use slices of spheroid? comment on the distribution of live/dead cells on inner parts.

(3) Some of the key relevant work has not been included in the references section. Please include-

Markway, B.D., Cho, H. & Johnstone, B. Hypoxia promotes redifferentiation and suppresses markers of hypertrophy and degeneration in both healthy and osteoarthritic chondrocytes. Arthritis Res Ther 15, R92 (2013). https://doi.org/10.1186/ar4272

Meretoja, Ville V et al. “The effect of hypoxia on the chondrogenic differentiation of co-cultured articular chondrocytes and mesenchymal stem cells in scaffolds.” Biomaterials vol. 34,17 (2013): 4266-73. doi:10.1016/j.biomaterials.2013.02.064

Yang Shi, Jingyun Ma, Xu Zhang, Hongjing Li, Lei Jiang, Jianhua Qin, Hypoxia combined with spheroid culture improves cartilage specific function in chondrocytes, Integrative Biology, Volume 7, Issue 3, March 2015, Pages 289–297, https://doi.org/10.1039/c4ib00273c

Foldager CB, Nielsen AB, Munir S, Ulrich-Vinther M, Søballe K, Bünger C, Lind M. Combined 3D and hypoxic culture improves cartilage-specific gene expression in human chondrocytes. Acta Orthop. 2011 Apr;82(2):234-40. doi: 10.3109/17453674.2011.566135

Minor concerns:

(1) Line 77- consider defining the abbreviation ACL reconstruction

(2) Line 82, section 2.2- As the manuscript focuses on the influence of hypoxia on chondrogenic phenotype maintenance, it is necessary to mention the precise oxygen concentration for normoxic culture conditions.

(3) Line 82, section 2.2- mention the references studied to devise the protocol

(4) For figure 1- sections c and d, use the same scale on Y-axis. It would be easier for the readers to comprehend.  

(5) The manuscript needs to be proofread for typos (eg. line 199 radio in place of ratio) and check the usage- hypoxia or hypoxic throughout the manuscript.

(6) Figure 2- please mark normoxic culture and hypoxic culture on the figure to make it easier to understand.

Author Response

Dear reviewer,

We appreciate your observations which have been addressed in the revised version improving the overall quality of our work.

Regarding minor concerns,

1) Line 77- consider defining the abbreviation ACL reconstruction

R: Thank you for your comments. We have defined the abbreviation ACL in line 77.

(2) Line 82, section 2.2- As the manuscript focuses on the influence of hypoxia on chondrogenic phenotype maintenance, it is necessary to mention the precise oxygen concentration for normoxic culture conditions.

R:  We appreciate your comment. In line 89, we have added the oxygen concentration for normoxic culture conditions (21% O2).

(3) Line 82, section 2.2- mention the references studied to devise the protocol

R: We appreciate your observation. Reference studied has been added to section 2.2.

(4) For figure 1- sections c and d, use the same scale on Y-axis. It would be easier for the readers to comprehend.  

R: We have added more information in figure 1 regarding percentage of cells with chondrocyte-like morphology as reviewer 1 recommended for a better understanding. We would like to mention that scale on Y-axis correspond to the same one in both sections C and D. We transformed our raw data to Log of Relative Expression Units for an easy visualization.

(5) The manuscript needs to be proofread for typos (eg. line 199 radio in place of ratio) and check the usage- hypoxia or hypoxic throughout the manuscript.

R: We appreciate your observations. Typos have been corrected and the manuscript was proofread.

(6) Figure 2- please mark normoxic culture and hypoxic culture on the figure to make it easier to understand.

R: We have added a mark for normoxic and hypoxic culture on the top of each section in figure 2.

Major concerns:

(1) Data represented in Figure 1- SOX9, COL1A1 expression levels are similar in normoxic and hypoxic cultures. ACAN expression data shows large variation, especially for hypoxic culture-P0. In all only COL2A1 and COL10A1 have some difference in expression. Looking at the p values, this experimentation is not very conclusive.

R: We appreciate your observations. It is important to mention that COL2A1 was down-regulated with increasing culture passages in normoxia which affected the COL2A1/COL1A1 ratio as we stated in line 189 to 204. This result represents a clear characteristic of de-diferentiation in normoxic culture in comparison to those cells under hypoxia. Moreover, we would like to mention that COL10A1, a major marker of chondrocyte hypertrophy, was up-regulated in normoxia compared to cells under hypoxia. In this sense, the ratio COL2A1/COL10A1 was clearly affected in a culture time-dependent manner in cells cultured in normoxia.

 (2) section 3.3.2 How was the viability assay performed- did the authors use slices of spheroid? comment on the distribution of live/dead cells on inner parts.

R: The viability assay was performed in spheroids at day 14. The protocol was carried out directly in agarose microwell containing spheroids and analyzed in an inverted fluorescence microscope (EVOS FL Auto) allowing us to detect the total fluorescence intensity. This microscope does not allow us to analyze in a proper manner the inner parts or spheroids. Therefore, we described areas of dead cells in a general way as well as a detection of reduced cell viability. Line 233 to 240. In this sense, we increased details of viability assay in section 2.6 for a better comprehension.

(3) Some of the key relevant work has not been included in the references section. Please include-

Markway, B.D., Cho, H. & Johnstone, B. Hypoxia promotes redifferentiation and suppresses markers of hypertrophy and degeneration in both healthy and osteoarthritic chondrocytes. Arthritis Res Ther 15, R92 (2013). https://doi.org/10.1186/ar4272

Meretoja, Ville V et al. “The effect of hypoxia on the chondrogenic differentiation of co-cultured articular chondrocytes and mesenchymal stem cells in scaffolds.” Biomaterials vol. 34,17 (2013): 4266-73. doi:10.1016/j.biomaterials.2013.02.064

Yang Shi, Jingyun Ma, Xu Zhang, Hongjing Li, Lei Jiang, Jianhua Qin, Hypoxia combined with spheroid culture improves cartilage specific function in chondrocytes, Integrative Biology, Volume 7, Issue 3, March 2015, Pages 289–297, https://doi.org/10.1039/c4ib00273c

Foldager CB, Nielsen AB, Munir S, Ulrich-Vinther M, Søballe K, Bünger C, Lind M. Combined 3D and hypoxic culture improves cartilage-specific gene expression in human chondrocytes. Acta Orthop. 2011 Apr;82(2):234-40. doi: 10.3109/17453674.2011.566135

R: We appreciate your observation. These relevant references have been used to increase the overall quality of the discussion related to 3D culture combined with hypoxia.

We would like to mention that we increased discussion regarding the role of hypoxia regulating gene expression in chondrocytes in 2D culture and scaffold-free spheroids. Line 323 – 331, 334 – 338, 350 – 380. We also rewrite our conclusion found in line 422 – 424.

Round 2

Reviewer 1 Report

Dear authors, thank you for the discussion and your work. I have no further concerns.

Author Response

We appreciate your observations, which have increased the overall quality of our manuscript.

Reviewer 2 Report

The overall quality of the manuscript is improved after revision.

Author Response

(The authors gave the same response as above.)

Reviewer 3 Report

The authors have revised the manuscript and addressed the comments.

Author Response

(The authors gave the same response as above.)
